# Electrolyte Analysis in Blood Serum by Laser-Induced Breakdown Spectroscopy Using a Portable Laser

**DOI:** 10.3390/molecules27196438

**Published:** 2022-09-29

**Authors:** Zhongqi Feng, Shuaishuai Li, Tianyu Gu, Xiaofei Zhou, Zixu Zhang, Zhifu Yang, Jiajia Hou, Jiangfeng Zhu, Dacheng Zhang

**Affiliations:** 1School of Optoelectronic Engineering, Xidian University, Xi’an 710071, China; zhongqifeng@stu.xidian.edu.cn (Z.F.); 20051212253@stu.xidian.edu.cn (S.L.); tygu@stu.xidian.edu.cn (T.G.); zhangzixv@foxmail.com (Z.Z.); houjj@xidian.edu.cn (J.H.); jfzhu@xidian.edu.cn (J.Z.); 2Clinical Laboratory, The Hospital of Xidian University, Xi’an 710071, China; zhouxfzyh712@163.com; 3Department of Pharmacy, Xijing Hospital, Xi’an 710032, China

**Keywords:** blood serum, electrolytes, laser-induced breakdown spectroscopy, partial least squares regression

## Abstract

The fast and reliable analysis of electrolytes such as K, Na, Ca in human blood serum has become an indispensable tool for diagnosing and preventing diseases. Laser-induced breakdown spectroscopy (LIBS) has been demonstrated as a powerful analytical technique on elements. To apply LIBS to the quantitative analysis of electrolyte elements in real time, a self-developed portable laser was used to measure blood serum samples supported by glass slides and filter paper in this work. The partial least squares regression (PLSR) method was employed for predicting the concentrations of K, Na, Ca from serum LIBS spectra. Great prediction accuracies with excellent linearity were obtained for the serum samples, both on glass slides and filter paper. For blood serum on glass slides, the prediction accuracies for K, Na, Ca were 1.45%, 0.61% and 3.80%. Moreover, for blood serum on filter paper, the corresponding prediction accuracies were 7.47%, 1.56% and 0.52%. The results show that LIBS using a portable laser with the assistance of PLSR can be used for accurate quantitative analysis of elements in blood serum in real time. This work reveals that the handheld LIBS instruments will be an excellent tool for real-time clinical practice.

## 1. Introduction

For the diagnosis of diseases or their prevention and control, millions of daily analyses are performed by clinical laboratories around the world [1]. Among these common analyses, the provision of fast and reliable data for electrolytes (K, Na, Ca) in blood serum is required. Due to their robustness and low cost, electrochemical methods or colorimetry/complexometric titration incorporated in biochemical analyzers have been widely used to determine the concentrations of serum electrolytes [1,2,3]. Atomic absorption spectroscopy (AAS) is also an accurate analysis method, but it has not been massively used in clinical practice due to its complex analysis process [4]. Inductively coupled plasma optical emission spectrometry (ICP-OES) and mass spectrometry (ICP-MS) provide sensitive methods for elemental analysis, but they need cumbersome sample preparation and cost too much [5,6]. Although many techniques for analyzing electrolytes are available, cheaper, faster and simpler methods are still demanded in real-time clinical practice.

Laser-induced breakdown spectroscopy (LIBS) has been demonstrated as a powerful elemental analysis technique on substances due to its multi-element analysis, high sensitivity and real-time process [7,8]. It allows fast contact-less analysis of any material and has unique versatility and capabilities for real-time composition determination [9,10]. At present, LIBS plays an important role in the detection of human tissues [11]. For application to blood, excellent work has also been performed by many researchers. In 1994, Cheung et al. reported the measurement of K and Na in human blood cells by LIBS. The relative standard deviation (RSD) of the signal was less than 9% [12]. Krum et al. used LIBS to measure the electrolytes in human blood serum, and the quantitative accuracy was better than 10% [1]. Khumaeni et al. used LIBS to identify elements in human blood serum. Elements including C, Ca, Na, H, O and molecular band CN were clearly observed in LIBS spectra [13]. Melikechi et al. compared the blood spectra with the spectra of other organic compounds [14]. A correct rate of more than 98% was obtained by chemometrics and machine learning [15]. Chu et al. used LIBS for the discrimination of nasopharyngeal carcinoma serum. The correct rate was better than 98% with the help of extreme learning machine classifier [16]. Berlo et al. applied LIBS and machine learning to distinguish serum of donors who previously tested positive for SARS-CoV-2 from those who did not, with an up to 95% correct rate [17]. Mohammad et al. used laser-induced fluorescence to enhance the LIBS signal of Rb in blood. The limit of detection (LOD) was down to 90 mg/L [18]. Our team has proposed a new method for liquid LIBS measurement. The LOD of Na was determined to be 1 mg/L in a Na_2_CrO_4_ solution [19].

From the work above, it can be found that LIBS is a potential technique for blood serum measurement. However, the application of LIBS in clinical practice is still limited by many factors, especially its size. In this work, a LIBS system equipped with a self-developed portable laser was used to analyze the electrolytes in blood serum, which could help to develop smaller-scale, lower-cost clinical equipment. The partial least squares regression (PLSR) analysis method was used to improve the accuracy of quantitative analysis for K, Na, Ca.

## 2. Experimental Setup and Sample Presentation

### 2.1. LIBS System

The experiments were carried out with the LIBS system shown in Figure 1. A self-developed portable Nd:YAG laser was used as the ablation light source, which could deliver 7 mJ pulse energy at its fundamental wavelength with an energy fluctuation of 0.8%. The laser pulse duration was smaller than 2 ns at 10 Hz repetition rate. The laser beam was focused on the sample using a quartz lens with 60 mm focal length. Plasma emission was focused on a bifurcated fiber cable by a pair of plano-convex lenses. The fiber was connected to a six-channel fiber optic spectrometer (AvaSpec Multi-Channel, Avantes, NLD, Apeldoorn, The Netherlands) with a spectral resolution of 0.08~0.11 nm in the range of 220~880 nm. The signals were recorded by CCD detectors with 2 ms minimum gate width. A photodiode detector was mounted behind the reflection mirror to monitor remaining laser pulses and trigger the spectrometer. The samples were stuck in a 360° motorized rotation stage to refresh the target point and avoid the destruction of samples. All of the experiments were carried out in air without any control of the surrounding atmosphere.

### 2.2. Sample Preparation

The samples were 53 individual human blood serum specimens provided by the Hospital of Xidian University. These samples were first deposited on qualitative filter paper (BWD 101, Baoweide Environmental Protection Technology Co., Ltd., Suzhou, China) and glass slides (Sail Brand 7101, Jiangsu Qipinsheng Medical Products Co., Ltd., Taizhou, China), respectively. The filter paper met the standard of GB/T 1914-2017. Then, the samples were dried for 10 min under laboratory conditions. Each sample on different supports was used only one time. To avoid interference from the supports, blank filter paper and glass slides were also measured and deducted from the spectra of samples. The spectrometer was triggered by the laser pulse, and the delay time between laser ignition and spectral acquisition was fixed at 1.3 μs. To improve the repeatability of measurements, 10 spectra were acquired for each sample, and each spectrum was an average result of 50 laser pulses.

## 3. Result and Discussion

### 3.1. Spectra Presentation

The LIBS spectra from the samples deposited on these two supports and the blank supports are presented in Figure 2. The characteristic lines of the target elements including Na, K, Ca and Si were observed in these spectra. By comparing Figure 2a,c, the line of Si 288.16 nm was only observed on the blank glass slide, but not in blood serum deposited on glass slides. This meant that there was not any interference with the blood serum spectrum from the glass slide support. However, the same characteristic lines from K, Na and Ca could be observed both in Figure 2b,d. This meant that the filter paper support could easily interfere with the spectrum of blood serum. The interference was also verified by inconsistency in the spectra of blood serum on different supports. Therefore, the filter paper as a support needed to be deducted from the spectra of samples in the following analysis.

### 3.2. Quantitative Analysis by PLSR

Chemometrics provides accurate quantitative methods for the analysis of LIBS spectra [20]. In this work, the partial least squares regression (PLSR) analysis method was used to build the quantitative models. PLSR is a linear regression algorithm based on multivariate analysis [21]. In PLSR, multicollinearity problems can be avoided by transforming initial input data into the latent variables (LVs) uncorrelated with each other and deriving regression coefficients [22].

The PLSR process was conducted using the statistical functions of Origin 2022 Learning Edition. Fifty-two arbitrary serum spectra were used as a training set, and the remaining one spectrum was used as test set. All of the lines from the normalized spectra were used as input data. The intensity of normalized characteristic lines was transformed into a matrix X, and the reference concentrations of K, Na, and Ca obtained with a commercial electrolyte analyzer were transformed into a matrix Y. Both X and Y in the training set were used to train the PLSR model. The PLSR model of blood serum supported by glass slides was established.

Figure 3 shows the description of variance explained by single LV and cumulative LVs for matrix X and Y. As shown in Figure 3a, the information of variance explained by the single LV had almost no contribution when the number of LVs exceeded 9. More than 90% of the variance information for matrix X was explained by the cumulative LVs. However, the variance information for matrix Y only increased to 80%, accumulating 12 LVs, as shown in Figure 3b. So, the number of LVs from 12 to 20 was optimized for the most accurate PLSR model.

Figure 4 shows the optimization of the number of LVs. The mean prediction errors of the test set were computed by the PLSR model with different numbers of LVs. It was found that the mean prediction error was declining with the increase in LVs from 12 to 16. At this range, more information available in LVs contributed to a more accurate result. However, the mean prediction error showed a sharp rise when the number of LVs was increased to 19. This may be due to the overfitting for the smaller groups of data in the training set. Therefore, the optimal parameter of PLSR was fixed at 16 LVs for its lower error and reduced computational burden.

The results of optimal PLSR are shown in Figure 5 for the samples deposited on glass slides. Both the training set and test set were predicted by the PLSR model, respectively. The reference concentrations of elements K, Na, and Ca were provided by a medical electrolyte analyzer (CBS-400), and the PLSR predicted concentrations were measured by LIBS. The correlation between reference concentration and PLSR-predicted concentration was revealed with the aid of reference curve y = x. In Figure 5, all of the points for the training set were close to the reference curve, showing great agreement between reference and predicted concentrations. This meant the PLSR model was reliable. For the test set points, the predicted concentrations of K, Na, and Ca were also highly consistent with the reference concentrations. As listed in Table 1, the prediction accuracies for K, Na, and Ca were 1.45%, 0.61% and 3.80%, respectively. Hence, the PLSR model was also verified to be accurate.

From the results of optimal PLSR for the samples deposited on glass slides, the accurate quantitative analysis of K, Na, and Ca was obtained. In the same way, the concentrations of K, Na, and Ca in blood serum deposited on filter paper were also predicted, as shown in Figure 6. Great linearity was also obtained between the reference and predicted concentrations for the training set and test set. As listed in Table 2, the prediction accuracies for K, Na, and Ca were 7.47%, 1.56% and 0.52%, respectively. Although the accuracy for K was poorer than that on glass slides, the improvement in accuracy for Ca was also a positive result. This meant that filter paper could also provide a good support for serum LIBS.

The results listed in Table 1 and Table 2 show that accurate analysis of electrolytes in blood serum can be realized by LIBS and PLSR. However, more work still needs to be carried out to improve the accuracies for Ca in blood serum deposited on glass slides and K in blood serum deposited on filter paper. For the serum deposited on glass slides, the substrate was not ablated by the laser. The poorer accuracy for Ca may be attributed to its lower concentration in blood serum, which was also verified by the lower intensity and stability of Ca lines. For the serum deposited on filter paper, serum and substrate were ablated by the laser at the same time. The approximately 34% attenuation for the K line at 766.49 nm due to the change in matrix may be one of the reasons for the negative variation in the results.

## 4. Conclusions

To conclude, LIBS with a portable laser was used for rapidly quantitative analysis of electrolyte elements including K, Na, and Ca in blood serum. Great linearity and prediction accuracy were obtained by the PLSR method for the serum samples both on glass slides and filter paper. The prediction results for K, Na, Ca, respectively, were 1.45%, 0.61% and 3.80% for serum on glass slides and 7.47%, 1.56% and 0.52% on filter paper. This means that LIBS using a portable laser meets the requirements of the detection index for human health. To report results on different supports, the effects of glass slides and filter paper on the predicted concentrations were also presented in this paper. In the LIBS measurement, it is necessary to select a suitable support according to the type and concentration of target element. The application of a portable laser for accurate quantitative analysis of serum also demonstrated the possibility of using handheld LIBS in real-time clinical practice. In the future, smaller components such as mini spectrometers will be used for LIBS measurements in real time.

## Figures and Tables

**Figure 1 molecules-27-06438-f001:**
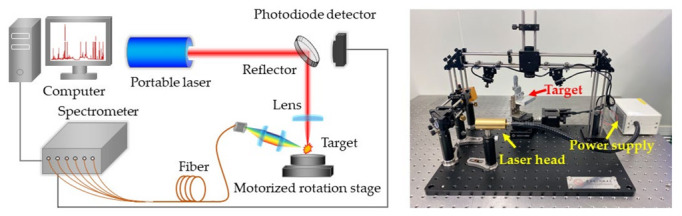
Schematic and picture of the LIBS experimental setup.

**Figure 2 molecules-27-06438-f002:**
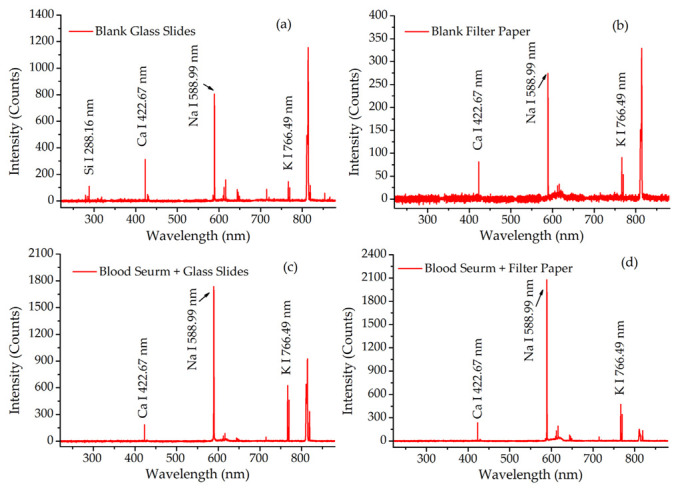
The LIBS spectra of samples and their blank supports. (**a**) Blank glass slides; (**b**) blank filter paper; (**c**) blood serum deposited on glass slides; (**d**) blood serum deposited on filter paper.

**Figure 3 molecules-27-06438-f003:**
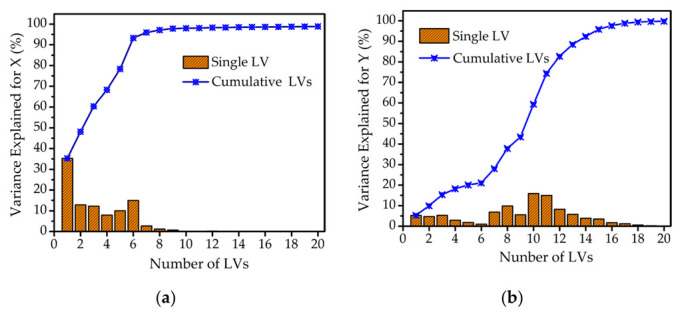
Description of the variance explained by single LV and cumulative LVs. (**a**) Variance explained for matrix X. (**b**) Variance explained for matrix Y.

**Figure 4 molecules-27-06438-f004:**
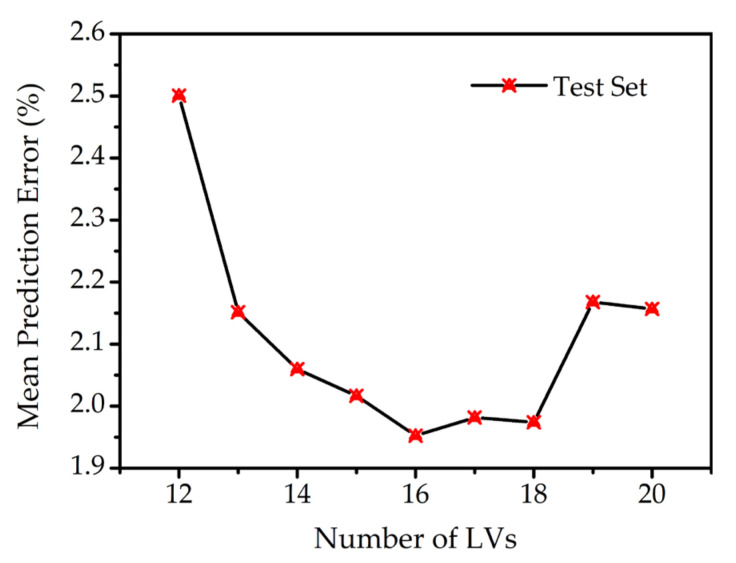
Mean prediction error with different numbers of LVs.

**Figure 5 molecules-27-06438-f005:**
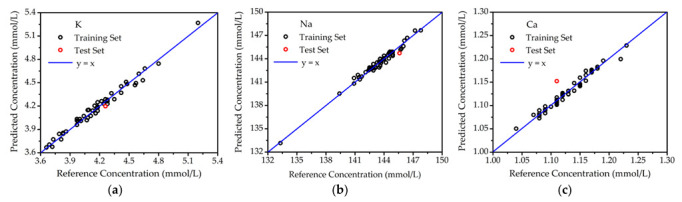
The quantitative analysis of elements in blood serum on glass slides. (**a**) K; (**b**) Na; (**c**) Ca.

**Figure 6 molecules-27-06438-f006:**
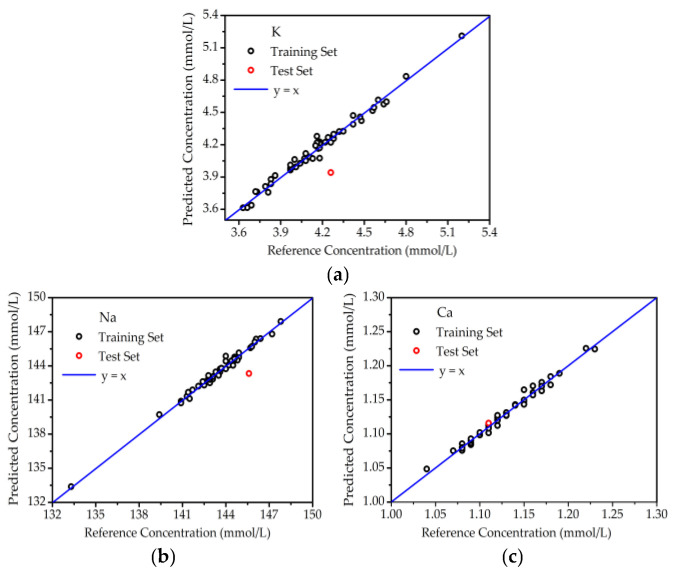
The quantitative analysis of elements in blood serum on filter paper. (**a**) K; (**b**) Na; (**c**) Ca.

**Table 1 molecules-27-06438-t001:** The results for blood serum on glass slides using the test set.

Target Element	Reference Concentration (mmol/L)	Predicted Concentration (mmol/L)	Accuracy (%)
K	4.26	4.20	1.45
Na	145.60	144.72	0.61
Ca	1.11	1.15	3.80

**Table 2 molecules-27-06438-t002:** The results for blood serum on filter paper using the test set.

Target Element	Reference Concentration (mmol/L)	Predicted Concentration (mmol/L)	Accuracy (%)
K	4.26	3.94	7.47
Na	145.60	143.33	1.56
Ca	1.11	1.12	0.52

## Data Availability

Not applicable.

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
