# Peer review of "Electrolyte Analysis in Blood Serum by Laser-Induced Breakdown Spectroscopy Using a Portable Laser"

_molecules, 2022, doi:10.3390/molecules27196438_

Round 1
Reviewer 1 Report
In summary, I accepted this paper for publication in this format. I suggest minor revision and the incorporation of all important points.
The authors have chosen a very nice and interesting topic for their work.
The use of principal component analysis (PCA) and hierarchical cluster analysis (HCA) for multivariate is very important for the statistical analyses. Please insert this technique in the manuscript.
Please! Check the English! There are few mistakes throughout the manuscript which need to be corrected as well as some simple grammatical errors to be ironed out.
Use the article “the” before the chemical formula throughout the manuscript.
Please format this paper! There are others mistakes to be formated in the manuscript .
Author Response
1.The use of principal component analysis (PCA) and hierarchical cluster analysis (HCA) for multivariate is very important for the statistical analyses. Please insert this technique in the manuscript.
Reply: Thanks for your suggestion. In our manuscript, the PLSR we used is a method of multivariate analysis. The PLSR latent variables (LVs) uncorrelated with each other were used to represent the initial input data, which has a similar function to the principal components in PCA. So, we do not need to use other multivariate analysis methods such as PCA and HCA.
2. Please! Check the English! There are few mistakes throughout the manuscript which need to be corrected as well as some simple grammatical errors to be ironed out.
Use the article “the” before the chemical formula throughout the manuscript.
Please format this paper! There are others mistakes to be formatted in the manuscript.
Reply: Thanks for your reminder. We have checked and corrected the mistakes in manuscript as far as possible.
Reviewer 2 Report
This work, when successful, will greatly reduce the cost and size of the system and improve the practicability of LIBS for the analysis of medical samples. This is a major issue that is often not addressed with the attention that it deserves particularly when quantitative information is sought from the data. For this reason, I find the subject matter addressed to be relevant to the field of spectroscopy (LIBS) and for the ever-growing attempts to develop devices and methods for quantification of medical sample.
However, I found that the manuscript lacks significant information (laser energy used, type of “medical” paper used) which makes it hard to follow. As a result, I find that the conclusions of the work are not clearly supported by the data.
1. Provide details of the measurements and in particular key parameters such as the laser energy, the focusing conditions, the type (details) of substrate used.
2. Of 53 samples available for this study, 52 were used as a training set and only one (1) sample as a test set (see section 3.2)! It begs the question why they did not do at least a leave one out approach? How would trying a different sample (as a test) impact Figs. 4-6? This is the biggest concern I have with this manuscript.
3. It is not obvious to this reviewer that self absorption is the main reason behind the quantitative results obtained for K, Na, and Ca (see table 1 and table 2) obtained with a glass slide and a “medical paper”. Consider discussing this point and consider providing details about the "medical" paper used.
4. Consider improving the English and double checking and updating the references.
I encourage you to work more on this important subject matter to improve the manuscript before resubmitting for a review in Molecules or any other scientific journal.
Author Response
However, I found that the manuscript lacks significant information (laser energy used, type of “medical” paper used) which makes it hard to follow. As a result, I find that the conclusions of the work are not clearly supported by the data.
1. Provide details of the measurements and in particular key parameters such as the laser energy, the focusing conditions, the type (details) of substrate used.
Reply: Thanks for your suggestion. The laser energy used in this work is 7 mJ with the energy fluctuation of 0.8%. The laser beam was focused by a quartz lens with 60 mm focal length in this work. These parameters have been added in section 2.1.
Two substrates, the glass slides (Sail Brand 7101, China) and qualitative filter paper (BWD 101, China), were used in this work. The blood serum was not penetrated by laser when deposited on glass slides (discussed in section 3.1). The filter paper meets the standard of GB/T 1914-2017. These parameters have been added in section 2.2.
Although the qualitative filter paper was used in our work, we deducted the spectrum of blank filter paper in the following quantitative analysis.
2. Of 53 samples available for this study, 52 were used as a training set and only one (1) sample as a test set (see section 3.2)! It begs the question why they did not do at least a leave one out approach? How would trying a different sample (as a test) impact Figs. 4-6? This is the biggest concern I have with this manuscript.
Reply (1): Thanks for your suggestion.
As we known, the leave one out approach is a great method for cross validation.
1) But, too much computation is required using leave one out method in PLSR modeling. In our PLSR process, we use any 52 samples as the training set and the remaining 1 as the test set in section 3.2.
2) To ensure the rationality of parameter optimization, we studied the initial data information contained in LVs, firstly. When the appropriate amount of information is included in the LVs, the predicted results can be more accurately close to the true value.
3) In the figure 3 and 4, within our LVs optimization range, the average prediction accuracy range varies in a small range (1.9~2.5%). Adopting leave one out approach only changes one training data once time. It has not great affect on the information contained in LVs.
4) For the 1 test set, if the measurement repeatability is great, the predicted results should be also similar to that in figure 5 and 6. 5) What’s more, we believe that more computing needs to be used to build a database with huge data. Using the method in our work can also obtain the better PLSR parameters and prediction results with less computation.
As discussed above, we think the leave one out approach is unnecessary in our work.
(2) For the question “How would trying a different sample (as a test) impact Figs. 4-6?”, we think that it has slight impact on the parameter optimization and prediction results. The reasons are presented as following.
1) Before performing parameter optimization, we determined the range of the LVs number that need to be optimized firstly. At this range, the information contained in the LVs is sufficient to support the establishment of the PLSR model. So, different test data does not have much effect on figure 4.
2) The repeatability of data is maintained by stability of portable laser and statistical methods. The portable laser can work continuously and stably for more than 3 hours at the energy of 7 mJ with energy fluctuation of 0.8%. In data processing, multiple spectra averaging and normalization can further improve data repeatability.
3). In PLSR process, we also computed the prediction results of the model on the training set data. In the figure 5 and 6, great linearity is obtained and the linearity is only affected by training set. All of data are well distributed closed to the line y=x. It also means that any one of the 53 data as a prediction will not have a big impact on the results.
When the conditions above are guaranteed, use different test set has slight impact on the results in our work.
3. It is not obvious to this reviewer that self-absorption is the main reason behind the quantitative results obtained for K, Na, and Ca (see table 1 and table 2) obtained with a glass slide and a “medical paper”. Consider discussing this point and consider providing details about the "medical" paper used.
Reply: Thanks for your suggestion. The element content of the filter paper was not provided by the manufacturer. After much consideration, we agreed your point, the results in table 1 and table 2 attributed to self-absorption was not stringent enough. Although there could be many causes for the results, we analyzed possible causes based on the information obtained in LIBS spectra. So, the relative statements have been revised as:
“From the results shown in table 1 and table 2, the accurate analysis of electrolytes in blood serum can be realized by LIBS and PLSR. However, more work still needs to be carried out to improve the accuracies for Ca in blood serum deposited on glass slides and K in blood serum deposited on filter paper. For the serum deposited on glass slides, the substrate was not ablated by laser. The poorer accuracy of Ca may be attributed to its lower concentration in blood serum, which was also verified by lower intensity and stability of Ca lines. For the serum deposited on filter paper, serum and substrate were ablated by laser at the same time. The approximately 34% attenuation for K line at 766.49 nm due to the change of matrix may be the one of causes for the negative variation in results.”
4. Consider improving the English and double checking and updating the references.
Reply: Thanks for your suggestions. We have improved the English as far as possible. The references also have been double checked and updated.
Round 2
Reviewer 2 Report
I find the manuscript to be much improved. The authors have provided some key elements that make the manuscript easier to follow. However, this manuscript still falls short of the level required for publication in a peer-reviewed journal but, I do believe, that the shortcomings can be fixed by the authors.
Below, I will list points that the authors need to address to enhance the quality of the manuscript.
1. English still requires some improvement.
a. Example, this is not a “self-developed” LIBS device but a homemade portable LIBS system,
b. Section 2.1: “the laser pulse duration is smaller than 2 ns”. Please provide how long the pulse is , is it 1 ns , microseconds? And provide if need be an uncertainty on this number. In addition, it is not “smaller” but shorter.
c. Plural of serum is sera (change everywhere where appropriate).
2. Section 3.2
a. Consider discussing in more details how the normalization was performed. Was it done with respect to the total intensity? If so, how?
b. What is the effect of background on the normalization?
c. The authors state “The intensity of normalized characteristic lines was transformed into a matrix X, and the reference concentrations of K, Na, Ca obtained by commercial electrolyte analyzer were transformed into a matrix Y.” Consider providing details on the commercial electrolyte analyzer used. The manuscript only states that it is a CBS-400
3. LIBS work on blood sera has been conducted by numerous groups. Many of these works are not referred to. In addition, many scholars have used univariate analysis techniques to quantify compositions of samples analysed using LIBS. Consider refereeing to key references; examples include:
· https://doi.org/10.1016/j.sab.2016.07.008
· https://doi.org/10.1016/j.sab.2017.11.016
· https://doi.org/10.1016/j.sab.2018.05.010
· https://doi.org/10.1016/j.sab.2014.03.014 (LIBS quantification)
· Reference #14. Correct the authors’ names by deleting “Cote and Priezzzhev”. It should read:
Melikechi, N.; Ding, H.; Rock, S.; Marcano O, A.; Connolly, D. Laser-induced breakdown 250 spectroscopy of whole blood and other liquid organic compounds. In Proceedings of the Conference on Optical Diagnostics and Sensing VIII, San Jose, US, 21-23.01.2008. 252